# Bone Targeting Agents in Patients with Metastatic Prostate Cancer: State of the Art

**DOI:** 10.3390/cancers13030546

**Published:** 2021-02-01

**Authors:** Veronica Mollica, Alessandro Rizzo, Matteo Rosellini, Andrea Marchetti, Angela Dalia Ricci, Alessia Cimadamore, Marina Scarpelli, Chiara Bonucci, Elisa Andrini, Costantino Errani, Matteo Santoni, Rodolfo Montironi, Francesco Massari

**Affiliations:** 1Division of Oncology, IRCCS Azienda Ospedaliero-Universitaria di Bologna, Via Albertoni 15, 40138 Bologna, Italy; veronica.mollica2@unibo.it (V.M.); alessandro.rizzo11@studio.unibo.it (A.R.); matteo.rosellini@studio.unibo.it (M.R.); andrea.marchetti12@studio.unibo.it (A.M.); angeladalia.ricci@studio.unibo.it (A.D.R.); chiara.bonucci@studio.unibo.it (C.B.); elisa.andrini2@studio.unibo.it (E.A.); 2Section of Pathological Anatomy, School of Medicine, United Hospitals, Polytechnic University of the Marche Region, 60121 Ancona, Italy; a.cimadamore@staff.univpm.it (A.C.); m.scarpelli@staff.univpm.it (M.S.); r.montironi@staff.univpm.it (R.M.); 3Department of Musculo–Skeletal Oncology, IRCCS Istituto Ortopedico Rizzoli, 40136 Bologna, Italy; costantino.errani@ior.it; 4Oncology Unit, Macerata Hospital, 62100 Macerata, Italy; mattymo@alice.it

**Keywords:** prostate cancer, bone metastasis, CRPC, HSPC, zoledronic acid, bone-targeted agents, denosumab

## Abstract

**Simple Summary:**

Over the disease course of metastatic prostate cancer, approximately the 90% of patients develops bone metastases, with bone involvement frequently leading to various skeletal complications including pathological fractures, spinal cord compression, and pain. Notably enough, the peculiar inclination of prostate cancer cells to seed the bone is considered an important cause of morbidity for prostate cancer patients. Recent years have witnessed the advent of several novel treatments for prostate cancer, and therapeutic paradigms are rapidly shifting. In this review, we aim at giving an overview of current knowledge on the relationship between prostate cancer and bone, especially focusing on the use of bone-targeted agents in this setting.

**Abstract:**

Bone health represents a major issue in castration-resistant prostate cancer (CRPC) patients with bone metastases; in fact, the frequently prolonged use of hormonal agents causes important modifications in physiological bone turnover and most of these men will develop skeletal-related events (SREs), including spinal cord compression, pathologic fractures and need for surgery or radiation to bone, which are estimated to occur in almost half of this patient population. In the last decade, several novel therapeutic options have entered into clinical practice of bone metastatic CRPC, with recent approval of enzalutamide and abiraterone acetate, cabazitaxel chemotherapy and radium-223, on the basis of survival benefit suggested by landmark Phase III trials assessing these agents in this setting. Conversely, although bone-targeted agents (BTAs)—such as the bisphosphonate zoledronic acid and the receptor activator of nuclear factor kappa-B (RANK) ligand inhibitor denosumab—are approved for the prevention of SREs, these compounds have not shown benefit in terms of overall survival. However, emerging evidence has suggested that the combination of BTAs and abiraterone acetate, enzalutamide and the radiopharmaceutical radium-223 could result in improved clinical outcomes and prolonged survival in bone metastatic CRPC. In this review, we will provide an overview on bone tropism of prostate cancer and on the role of BTAs in metastatic hormone-sensitive and castration-resistant prostate cancer.

## 1. Introduction

Prostate cancer (PCa) represents the most commonly diagnosed tumors in the male population and is still one of the main causes of cancer-related death worldwide [1]. The natural history of PCa may potentially go through different phases, from a hormone-sensitive prostate cancer (HSPC) to a castration-resistant prostate cancer (CRPC) state in around 18–24 months, and from localized disease to nonmetastatic biochemical recurrence and, finally, to the presence of distant metastases [2]. Skeletal involvement is present in up to 90% of cases of metastatic PCa due to a peculiar tropism for the bone exhibited by this tumor [3]. For this reason, bone health management is one of the main issues to address when treating PCa patients.

Several anatomic, biological and molecular characteristics of this type of tumor have been called into account to explain its bone tropism [4].

The backbone of PCa treatment is androgen deprivation therapy (ADT), that, despite the benefits obtained in these patients, is associated with relevant side effects correlated to androgen suppression and consequent body mass changes [5]. In fact, ADT leads to an alteration in the body composition consisting in an increase of subcutaneous fat and a decrease of lean body mass [6,7]. Furthermore, ADT is associated with a prevalence of osteopaenia of 39% and osteoporosis of 41% in newly diagnosed PCa with an absolute bone mineral density loss of 5% in the first year of ADT due to the decreased levels of testosterone and estradiol, which regulate bone remodeling [8,9,10]. This process leads to a higher bone turnover and bone loss [11].

Bone-targeted agents (BTAs), like the bisphosphonate zoledronic acid (ZA) and the receptor activator of nuclear factor κ B ligand (RANKL) inhibitor denosumab, are approved drugs in the treatment scenario of CRPC with bone metastases for the prevention of skeletal-related events (SREs). SREs include pathologic fracture, an oncologic emergency such as spinal cord compression, need for surgery or radiation therapy to bone, hypercalcemia [12]. SREs are correlated to a decreased quality of life, increased pain, and shortened survival in PCa patients [13]. Notably enough, ZA has been suggested to play direct and indirect effects, inhibiting osteoblast-mediated bone resorption, tumor-associated angiogenesis, and tumor-self seeding [7,12]; in addition, ZA has been shown to play a synergistic effect when used in combination with cytotoxic drugs [8].

Differently, in the metastatic hormone-sensitive setting, several randomized clinical trials (RCTs) of BTAs did not show a benefit of these agents in term of time to first SRE, so they are not recommended for metastatic HSPC patients [14].

Herein, we provide an overview on literature focusing on the main mechanisms of bone tropism of PCa and we reported the results of the pivotal studies investigating BTAs in metastatic HSPC and CRPC.

## 2. Bone Metastasis in Prostate Cancer: The Underlying Molecular Mechanisms

Bone involvement in prostate cancer is an extremely frequent event, which is deemed to occur in up to 90% of patients with advanced disease [2]. From an anatomical point of view, the vertebral venous plexus—in which the prostatic venous plexus indirectly drains—has been suggested to be an important route of prostate cancer bone metastasization, due to its localization throughout the column [3]. However, several other nonanatomical elements have been observed to favor the onset of bone metastases, including the biological and molecular features of prostate tumor cells [4].

Several mechanisms have been reported to play a role in prostate cancer bone metastases; from a molecular point of view, homing of cancer cells to the bone represents a key concept [15]. Firstly, following epithelial-mesenchymal transition (EMT), prostate cancer cells acquire the capacity of departing from the primary site, thus emptying into blood circulation [16]. Subsequently, in order to invade bone sites, the adhesion between bone marrow endothelial cells (BMECs) and tumor cells is crucial, with E-selectin ligand playing an important role in this interaction. Additionally, several other adhesion molecules are involved [17]. Among these, the CD44—a multifunction cell surface adhesion receptor that is expressed by prostate cancer cells—binds to vascular cell adhesion molecule 1 (VCAM-1), and other factors, including insulin-like growth factor 1 (IGF1), insulin, and interleukin-17 (IL-17) are able to boost this process [18]. Moreover, the interaction between endothelial cells and homing of cancer cells to the bone in this malignancy includes the involvement of the integrin α_V_β_3_ [19]. The expression of this vitronectin receptor, which has been suggested to induce bone metastasization in other solid tumors, is increased by the stromal cell-derived factor 1 (SDF1), also known as C–X–C motif chemokine ligand 12 (CXCL12) [20]. Of note, the secretion of SDF1 depends on osteoblasts and bone marrow stromal cells, highlighting that the bone tropism of prostate cancer could be closely linked to signals from the bone marrow microenvironment [21]. Besides, SDF1 represents an important element in the regulation of hematopoietic stem cell (HSC) homing to the bone marrow, with previous reports suggesting that HSC homing has many similarities with bone metastasization mechanisms in prostate cancer, due to the analogous expression of several chemokines—including SDF1, CXCR7, and CXCR4 [22]. Another important element promoting prostate cancer bone metastases is CXCL-16, whose expression on prostate cancer cells led to the recruitment of mesenchymal stem cells (MSCs) from bone marrow, which express the CXCR6 chemokine receptor [23]. Of note, the interaction between CXCR6 and CXCL-16 hesitates in dissemination to the bone tissue through the transition of MSCs in cancer-associated fibroblasts [24]. In particular, cancer-associated fibroblasts are associated with high expression of SDF1, which induces the process of EMT in prostate cancer cells [25].

Intriguingly, after prostate cancer cells “nest” in bone tissue, these cells exploit the bone microenvironment [26]; here, cancer cells compete with HSCs in order to occupy this “niche”, lastly hesitating in bone colonization [27,28]. Following this process, prostate cancer tumor cells adapt to bone microenvironment in order to permit their survival, and tumor cells exploit mechanisms of immune system escape achieving a balance between apoptotic and proliferation processes [29,30]. In fact, once established in these niches, prostate cancer tumor cells remain in a dormant state. This stage has been properly called “immune-mediated dormancy” (or “population-level tumor dormancy”), and several molecular mechanisms have been suggested to play a role in this dormant phase [31]. For example, a wide number of reports observed that phosphorylated p38 and phosphorylated Erk may play an important role in immune-mediated dormancy, where higher levels of p38 lead cancer cells towards this phase [32].

Certainly, the following step is represented by the “exit” from the dormancy phase, resulting in the dissemination of prostate cancer cells [33]. Of note, despite prostate cancer bone metastases are commonly deemed to be osteoblastic, osteoclastic response plays an important role, since it has been associated with SREs and metastatic spread [34,35]. Due to the fact that bone is a dynamic organ receiving continuous remodeling, the interaction between osteoblastic and osteoclastic activities is strongly affected by the dissemination of cancer cells in the bone marrow environment [36]. In this process, RANKL—that is primarily expressed by stromal cells, osteocytes, and osteoblasts—has a leading role since the bond between RANK receptor and its ligand on osteoclast progenitors hesitates in bone resorption through osteoclastic differentiation [37,38] (Figure 1). Concurrently, the secretion of osteoprotegerin (OPG) by stromal cells and mature osteoblasts aims at preventing the RANK/RANKL association, by counteracting the osteoclastic activity [39].

These physiological mechanisms are heavily impaired in presence of bone dissemination. In fact, in the bone metastatic setting, RANKL expression is boosted by prostate cancer cells through the expression of parathyroid hormone-related protein (PTHrP), and in parallel, bone resorption is modified by the expression of matrix metalloproteinase-7 (MMP7) by osteoclasts [40,41]. These modifications are considered crucial for bone metastatic outgrowth, given that these aberrant activities lead to the replacement of physiological tissue with metastatic cells within the bone marrow, which have the place to grow and to “expand” freely [42]. Additionally, several studies have reported that prostate cancer cells express the Wnt inhibitor Dickkopf-1 (DKK-1) [43], and since Wnt regulates osteoblastic differentiation, the hyperexpression of DKK-1 results in the inhibition of osteoblastogenesis processes, further promoting the growth of prostate cancer cells within the bone marrow [44] (Figure 1). In addition, MSCs have been also suggested to sustain PCa growth by differentiating in adipocyte lineage. In particular, according to this interesting hypothesis, PCa cells localize to lipid-rich regions in bone marrow metastases, where they interact with adipocytes leading to increased proliferation [45]. The homing to the bone could be initially helped by a direct attraction by lipids released from bone marrow adipocytes. Moreover, selective lipids have been shown to be key factors for the progression of PCa cells, since these elements could act as pleiotropic factors able to stimulate gene expression, proliferation, and chemotaxis [46].

## 3. Bone Targeting Agents in Metastatic HSPC

Several clinical trials in metastatic HSPC (mHSPC) have recently investigated BTAs [47,48,49,50]. The phase III CALGB 90202 study assessed the time to first SRE and overall survival (OS) in 645 men randomized in a blinded 1:1 ratio to receive early treatment with ZA (4 mg/5 mL intravenously every four weeks) or placebo. This study highlighted no benefit for ZA in terms of median time to first SRE, that was 31.9 months in the experimental group (95% confidence [CI], 24.2 to 40.3) and 29.8 months in the placebo group (95% CI, 25.3 to 37.2; hazard ratio [HR], 0.97; 95% CI, 0 to 1.17; one-sided stratified log-rank *p*= 0.39) [48]. Moreover, OS was comparable in the two arms (HR, 0.88; 95% CI, 0.70 to 1.12; *p*= 0.29), and, in addition, the two groups presented similar rates of adverse events (AEs).

Another landmark study for BTAs in mHSPC is the STAMPEDE trial, a randomized controlled phase III study with a multiarm and multistage platform design [49]. In this study, time to first SRE was improved with the combination of docetaxel + ADT (112 patients reported SRE; HR 0.60; 95% CI, 0.48–0.74; *p* = 0.127 × 10^−5^) and docetaxel + ZA + ADT (108 patients; HR 0.55, 95% CI, 0.44–0.69; *p* = 0.277 × 10^−7^), but not with ADT + ZA (153 patients; HR 0.89, 95% CI 0.73–1.07; *p* = 0.221) or ADT alone. Median OS was 71 months for ADT alone, not reached for ADT + ZA (HR 0.94, 95% CI 0.79–1.11; *p* = 0.450), 81 months for ADT + docetaxel (HR 0.78, 95% CI 0.66–0.93; *p* = 0.006), and 76 months for standard of care (SOC) + ZA + docetaxel (HR 0.82, 95% CI 0.69–0.97; *p* = 0.022). These results suggested that ZA is not associated with benefits in terms of SREs, failure-free survival or OS in mHSPC patients.

The open-label phase ZAPCA III trial randomly assigned 227 men with mHSPC and bone metastases to combined ADT or ZA + ADT to demonstrate the superiority of the combination therapy in time to treatment failure (TTTF), time to the first SRE and OS [50]. Concordant to the results of the previous two studies, the ZAPCA failed to reach these endpoints. In particular, median TTTF were 12.4 months in the ADT + ZA arm and 9.7 months in the ADT alone arm (HR 0.75; 95% CI, 0.57–1.00; *p* = 0.051); in addition, median time to first SRE was 64.7 months and 45.9 months in mHSPCs receiving ADT + ZA and ADT alone, respectively (HR 0.58; 95% CI, 0.38–0.88; *p* = 0.009). The only exception was a subgroup of men with baseline PSA < 200 ng/mL treated with ADT + ZA, in which a significant delay of TTTF was detected: median TTTFs were 23.7 and 9.8 months for the ADT + ZA and ADT alone groups, respectively (HR 0.58; 95% CI, 0.35–0.93; *p* = 0.023).

It has to be underlined that denosumab (120 mg subcutaneous, monthly) has not been investigated in the hormone-sensitive setting and should not be used for this indication [47,51].

In conclusion, these results do not support an early treatment with ZA in mHSPC before the development of castration-resistant disease and, consequently, current guidelines do not suggest any BTA in clinical practice. Clinical trials investigating BTAs in mHSPC patients are reported in Table 1.

## 4. Bone Targeting Agents in Metastatic CRPC

In about 80% of metastatic PCa patients there is a relevant deterioration of quality of life due to skeletal affection and cancer treatment-induced bone loss. The management of bone metastasis plays a key role in nowadays clinical practice in CRPC patients with bone metastases [51,52,53,54,55].

As for BTAs, it has been showed that ZA reduces the incidence of SREs in men with CRPC with bone metastases and its combination with systemic oncologic treatments represents the standard-of-care in these patients [51]. Furthermore, denosumab has been shown to prevent SREs in the same setting as well.

The phase III, randomized and double-blind COU-AA-302 trial compared abiraterone acetate + prednisone with placebo + prednisone in a heterogeneous population of metastatic CRPC patients. Saad et al. published results of a post hoc analysis of this trial in 2015, which showed that in the subgroup of chemotherapy-naive patients the addition of BTAs, compared with no BTAs, was correlated to benefits in the experimental arm (abiraterone acetate + prednisone + BTA) with regard to primary endpoints (OS, time to Eastern Cooperative Oncology Group—ECOG—deterioration and time to opiate use for cancer-related pain) [52].

The increased effectiveness of ZA compared to other BP in reducing the incidence of SREs was demonstrated in the Zometa 039 phase III trial [55]. While other compounds, such as clodronate or pamidronate, did not achieve the outcome of delivering benefits in this setting [47,53,54]: ZA compared with placebo reported fewer SREs at 15 months’ follow-up (33.2% versus 44.2%; *p* = 0.021). Time to first SRE was also improved with ZA. Therefore, the only BP approved for CRPC with bone metastases is zoledronic acid [55].

Subsequent RCTs demonstrated the benefit of denosumab, a RANK-L inhibitor, in CRPC patients with bone metastases. In terms of mechanism of action, this agent binds the cytokine RANKL preventing it from binding the RANK receptor, and thus, denosumab blocks the maturation of osteoclast precursors, also promoting the apoptosis of mature osteoclasts [56] Denosumab (120 mg subcutaneous, every four weeks) has been shown superiority over ZA in improving time to first SRE in this setting [56]. Indeed, in 2011, Fizazi et al. published the results of a phase III study comparing denosumab with ZA in 1901 men with metastatic CRPC (Denosumab 103) [56]. This trial showed a significant benefit in terms of median time to first SRE in the denosumab arm compared to ZA arm (20.7 versus 17.1 months; *p* = 0.001 for noninferiority; *p* = 0.008 for superiority). As for AEs, patients in denosumab arm were more frequently affected by hypocalcemia (12.8% versus 5.8%) and by osteonecrosis of the jaw (ONJ) (ONJ; 2.3% versus 1.3%; *p* = 0.09, not statistically significant). These events are due to increased osteoclastic inhibition induced by denosumab.

Another aspect to take into consideration when choosing a BTA is that denosumab does not require dose adjustment in case of renal failure because its clearance does not depend on the kidney function. On the contrary, zoledronic acid’s dose adjustment is necessary in case of glomerular filtration rate (eGFR) 30 to 60 mg/min/1.7 m^2^ and its use is not recommended in case of eGFR < 30 mg/min/1.7 m2 [47,56].

Renal toxicity and the consequent possibility of skipping or delaying the administration of ZA may explain the longer duration of persistent therapy in patients using denosumab and the fact that these patients switch to zoledronic acid less frequently than vice versa, as emerged in a study that examined long-term treatment patterns of these two main BTAs [57].

Other bone-targeted agents are radiopharmaceuticals: among these, radium-223 (^223^Ra), an alpha-emitting agent, has a significant role in preventing symptomatic skeletal metastases from PCa. Radium-223 has been tested in the ALSYMPCA trial [58]. OS, the primary endpoint, and time to the first symptomatic SRE (one of the secondary end points) were consistently improved in the radium-223 group compared to placebo. Furthermore, radium-223 was associated with few AEs and a low rate of myelotoxicity [58].

The post hoc analyses of ERA-223 (a randomized, double-blind, placebo-controlled, phase III trial) evaluated the use of bisphosphonates or denosumab in chemotherapy naïve metastatic CRPC patients treated with abiraterone (and prednisone/prednisolone) + radium-223 or placebo [59]. In the subgroup of patients who received these BTAs at baseline the incidence of fractures was 15% in the radium-223 arm and 7% in the placebo one, compared to 37% and 15% in patients bone-health-agents-naive, respectively. These results led to the conclusion that the combination of abiraterone + prednisone + Radium-223 should not be used in first-line therapy for CRPC with bone metastases [59].

Moreover, the interim results of EORTC 1333/PEACE phase III study compared the combination of radium-223 + enzalutamide to enzalutamide alone in early asymptomatic or mildly symptomatic men with skeletal metastatic CRPC [60]. The design of the study has been modified consequently to the results of the ERA-223 trial, in order to coadminister BTAs such as bisphosphonates or denosumab. According to these early data, the cumulative risk of fracture at 13 months follow up was 12.4% in men treated with enzalutamide alone and 37.4% when radium-223 was added to enzalutamide. Anyway, the addition of BTAs to both arms showed a decrease in the 13-month cumulative risk of fracture close to 0% [60].

In conclusion, the results from ERA-223 and EORTC 1333/PEACE studies point out that improved prevention of pathological fractures is present only when BTAs are coadministered with combinations of radium-223 and novel hormone therapies [51,60].

The randomized controlled trial TRAPEZE tested docetaxel for six cycles + prednisolone with ZA, Strontium-89 (Sr-89) or both [61]. Sr-89, but not ZA, was associated with improved clinical progression-free survival (time to pain progression, SRE or death). Concerning ZA, this trial suggested a role in the post-chemotherapy maintenance therapy since emerged an improvement in SRE-free interval, mainly post disease progression. It is important to underline that both these agents improved OS [61].

Altogether these RCTs have demonstrated the overall benefits linked to BTAs in CRPC with bone metastases setting, especially in terms of increased time to the first SRE or reduced incidence of SREs. Nevertheless, several data reveal how a considerable part of men with metastatic CRPC do not receive an adequate treatment with ZA or denosumab in order to prevent SREs in clinical practice. According to a European study, 26% of patients with bone metastases did not receive a bone-targeting agent and only 53% received treatment within three months of bone metastases diagnosis [62]. Oncologists not only prescribed BTAs more frequently than urologists (78% vs 60%, respectively) but especially initiated treatment within three months of bone metastases diagnosis (56% vs 43%) [62].

Osteoclast inhibition might prevent bone metastases, according to preclinical data [63]. Based on this assumption, several studies have been designed in order to evaluate the hypothetical key role of bone target agents in skeletal metastases prevention in PCa clinical setting [63]. In the ZEUS trial, men with early stage high-risk PCa were treated with bisphosphonates (such as ZA or clodronate) even if there was not clinical evidence of skeletal affection at enrollment [64]: the results showed no benefits in terms of disease recurrence or bone metastases prevention due to BTA [64].

On the contrary, the use of denosumab in patients with biochemical recurrence but no radiological signs of bone metastases, was able to delay the time to symptomatic bone metastases, although OS was not improved [65]. Unfortunately, this data does not support the use of this drug in prophylaxis due to the 5% cumulative incidence of ONJ in the prolonged treatment (i.e., five-year schedule) [63,65].

Clinical trials investigating bone-targeting agents in mCRPC patients are reported in Table 2.

## 5. Conclusions

Bone metastases are an extremely frequent event in PCa, that influence prognosis and quality of life of these patients, thus making their management one of the principal issue in the treatment algorithm of this tumor. BTAs are pivotal treatments to integrate with oncologic therapies to improve patients’ outcomes in the metastatic castration-resistant setting, while they are not recommended in metastatic hormone-sensitive disease. Understanding the biological and molecular mechanisms underlying the bone tropism of PCa—including the possible role of inflammation and its association with tumor microenvironment [66]—could be helpful to open new treatment options and to guide the clinical management of these patients.

## Figures and Tables

**Figure 1 cancers-13-00546-f001:**
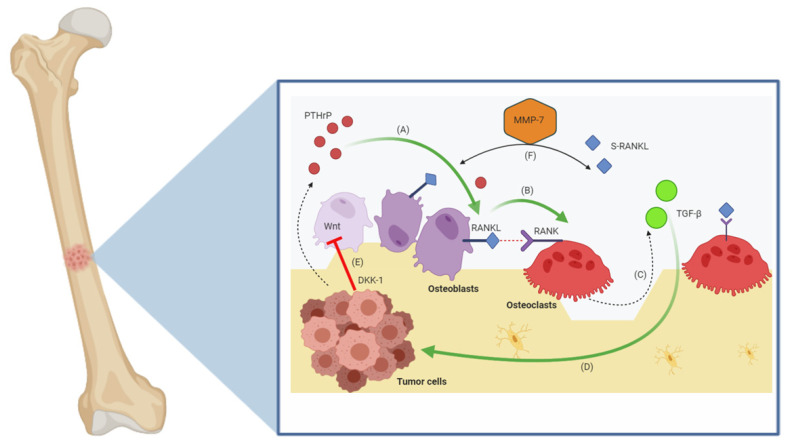
“Vicious cycle” in bone metastases. Bone metastatic prostate cancer cells secrete factors (e.g., parathyroid hormone-related protein, PTHrP) that upregulate receptor activator of nuclear factor kappa-B ligand (RANKL) expression by osteoblasts (**A**), which interacts with RANK receptor on osteoclast progenitors causing bone resorption through osteoclastic differentiation (**B**). This interaction consequently contributes to the release and activation of transforming growth factor-β (TGFβ) (**C**), which is stored in mineralized bone matrix and further promote tumor growth (**D**), leading to a “vicious cycle”. In addition, prostate cancer cells express the Wnt inhibitor Dickkopf-1 (DKK-1) and osteoblast progenitors respond to this negative regulator via inhibition of osteoblastogenesis process (**E**). Finally, the secretion of MMP-7 results in the solubilization of RANKL in the bone microenvironment (**F**), which increases bone resorption.

**Table 1 cancers-13-00546-t001:** Pivotal clinical trials investigating bone targeting agents in mHSPC.

Trial	Reference	Population (Number of Patients)	Treatment Arms	Primary End-Points	Secondary End-Points	Results
CALGB 90202 (Alliance)	Smith et al. [48]	mHSPC (645)	ZA vs. Placebo	Time to First SRE	OS	Time to first SRE: ZA: 31.9 months Placebo: 29.8 months HR 0.97; 95% CI, 0 to 1.17; one-sided stratified log-rank *p* = 0.39. Adjusted HR 0.88; 95% CI, 0.70 to 1.12; stratified log-rank *p* = 0.29
STAMPEDE	James et al. [49]	mHSPC: 61% M+ 15% *n* + /x M0 24% N0 M0 6% previously treated with local therapy (2962)	Arm A (ADT alone) vs. ArmB (ADT + ZA) vs. Arm C (Docetaxel + ADT) vs. Arm E (Docetaxel + ADT + ZA)	OS	TTfSRE	SREs reported in: Arm A: 328 pts Arm B: 153 pts HR 0.89, 95% CI 0.73–1.07; *p* = 0.221 Arm C: 112 pts HR 0.60, 95% CI 0.48–0.74; *p* = 0.127 × 10^−5^ Arm E: 108 pts HR 0.55, 95% CI 0.44–0.69; *p* = 0.277 × 10^−7^
ZAPCA	Kamba et al. [50]	mHSPC (227)	CAB vs. CAB + ZA	TTTF	TTfSRE OS	Median TTTFs: CAB + ZA: 12.4 months CAB: 9.7 months HR 0.75; 95% CI 0.57–1.00; *p* = 0.051 Median TTfSREs: CAB + ZA: 64.7 months CAB: 45.9 months HR 0.58; 95% CI 0.38–0.88; *p* = 0.009 OS was similar between the two groups.

ADT = androgen deprivation therapy; CAB = combined androgen blockade; CI = confidence interval; HR = hazard ratio; mHSPC = metastatic hormone-sensitive prostate cancer; OS = overall survival; pts = patients; SREs = skeletal-related events; TTfSRE = time to the first SRE; TTTF = time to treatment failure; vs. = versus; ZA = zoledronic acid.

**Table 2 cancers-13-00546-t002:** Pivotal clinical trials investigating bone targeting agents in mCRPC patients.

Trial	Reference	Population(Number of Patients)	Treatment Arms	Primary End-Points	Secondary End-Points	Results
COU-AA-302 (post-hoc)	Saad et al. [52]	mCRPC (no visceral metastases) and chemotherapy naive (1088)	BTAs + AAP vs. BTAs + placebo + prednisone	OS Time to ECOG deterioration Time to opiate use for CRP	-	BTT use showed significantly longer OS (*p* = 0.012; risk reduction 25%), longer time to deterioration in ECOG PS (*p* < 0.001, risk reduction 25%), and longer time to opiate use for CRP (*p* = 0.036, risk reduction 20%)
Zometa 039	Saad et al. [55]	mCRPC (643)	ZA Q4 W (4 mg or 8 mg) + ADT vs. Placebo + ADT	SRE–free survival; time to first SRE	-	ZA 4 mg was associated with fewer SRE (44.2% vs. 33.2%; difference = −11.0%, 95% CI −20.3% to −1.8%; *p* = 0.021) and increased median time to the first SRE (*p* = 0.011) vs. placebo
Denosumab 103	Fizazi et al. [56]	mCRPC Failure of at least one hormonal therapy as evidenced by a rising PSA (1901)	ZA Q4 W 4 mg IV and denosumab sc Q4 W 120 mg placebo vs. Denosumab sc Q4 W 120 mg and ZA 4 mg placebo IV	Time to first on-study SRE (noninferiority)	Time to first on-study SRE (superiority); time to first and subsequent on-study SRE	Median time to first on-study SRE: Denosumab: 20.7 months ZA: 17.1 months HR 0.82, 95% CI 0.71–0.95; *p* = 0.0002 for noninferiority; *p* = 0.008 for superiority
ALSYMPCA	Parker et al. [58]	mCRPC (921)	Radium-223 (6 injections at a dose of 50 kBq/Kg IV Q4 W) vs. Placebo	OS	Time to the first skeletal event	Median OS: Radium-223: 14.9 months Placebo: 11.3 months HR 0.70; 95% CI 0.58 to 0.83; *p* < 0.001 Median time to first symptomatic skeletal event: Radium-223: 15.6 months Placebo: 9.8 months HR 0.66; 95% CI 0.52 to 0.83; *p* < 0.001
ERA-223	Smith et al. [59]	mCRPC chemotherapy naive, asymptomatic or paucisymptomatic pts (806)	Radium-223 + AAP vs. Placebo + AAP	Symptomatic SRE-free survival	-	Median symptomatic SRE-free survival: Radium-223 + AAP: 22.3 months Placebo + AAP: 26.0 months HR 1.122; 95% CI 0.917 − 1.374; *p* = 0.2636
ERA-223 (post-hoc)	Smith et al. [59]	-	Radium-223 + AAP vs. Placebo + AAP vs. Radium-223 + AAP + BTAs vs. Placebo + AAP + BTAs	Incidence of pathological fractures	-	Incidence of pathological fractures: with BTAs: 15% in the radium-223 arm and 7% in the placebo arm without BTAs: 37% radium-223 arm and 15% in the placebo arm
EORTC 1333/PEACE III(safety analysis)	Tombal et al. [60]	mCRPC (146)	Enzalutamide vs. Enzalutamide + Radium-223 vs. Enzalutamide + BTAsvs. Enzalutamide + Radium-223 + BTAs	Fracture rate	-	Cumulative risk of fracture at a 13 months follow-up: without BTAs: 12.4% in enzalutamide arm vs. 37.4% in radium-223 arm with BTAs: 0% in enzalutamide arm vs. 2.2% in enzalutamide + radium-223 arm
TRAPEZE	James et al. [61]	mCRPC (757)	Docetaxel + prednisolone + ZA vs. Docetaxel + prednisolone + Sr-89 vs. Docetaxel + prednisolone + ZA + Sr-89	CPFS	SREFI OS	Sr-89 improved CPFS ZA did not improve CPFS but significantly improved SREFI HR 0.76; 95% CI 0.63 to 0.93; *p* = 0.008) Neither agent affected OS (Sr-89, *p* = 0.74; ZA, *p* = 0.91)
ZEUS (Zometa European Study)	Wirth et al. [64]	M0 prostate cancer At least one of the following: PSA ≥ 20 ng/mL; pN + disease; Gleason score 8–10.(1393)	SOC + ZA 4 mg/5 mL IV every 3 months (for a total of 48 months) vs. SOC only	Proportion of pts who develop BM during the study	Time to first BM OS Time to PSA doubling Safety Bone mineral density Biochemical markers of bone turnover	BIP–BM developed in: 88 of 515 patients (17.1%) in the ZA group 89 of 525 patients (17.0%) in the control group Chi-square test: *p* = 0.95

AAP = abiraterone acetate + prednisone; ADT = androgen deprivation therapy; BIP-BM = bone metastases diagnosed with bone-imaging procedures; BM = bone metastases; BTAs = bone-targeting agents; CI: confidence interval; CPFS = clinical progression-free survival; CRP = cancer related pain; ECOG PS = Eastern Cooperative Oncology Group Performance Status; HR= hazard ratio; mCRPC = metastatic castration-resistant prostate cancer; OS = overall survival; 4 QW = every 4 weeks; pts = patients; SOC = standard of care; SRE = skeletal-related event; SREFI = SRE-free interval; Sr-89 = strontium-89; vs. = versus; ZA = zoledronic acid.

## Data Availability

No new data were created or analyzed in this study. Data sharing is not applicable to this article.

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
