# Peer review of "Bone Targeting Agents in Patients with Metastatic Prostate Cancer: State of the Art"

_cancers, 2021, doi:10.3390/cancers13030546_

Round 1
Reviewer 1 Report
Skeletal related events (SREs) such as spinal cord compression and pathologic fractures often occur in men with prostate cancer. Main cause of SREs are treatment-related osteoporosis and bone metastasis, and SREs are correlated to a a decreased QOL and shorted survival in prostate cancer patients. Although bone-targeted agents (BTAs) including bisphosphonates and the antibody against RANKL are approved for the prevention of SREs, it is unclear whether the treatment with BTAs result in prolonged survival. In this article, the authors review the studies on the effectiveness of BTAs in metastatic prostate cancer.
Line 36; Ref.1 may be inappropriate because this article provides the cancer statics in the United States.
Line 64; Please add appropriate references for this sentence (Line 62-64).
Line 80; I think that Ref.15 is suitable for the next sentence (Line 80-82). Please add appropriate references for the sentence (Line 78-80).
Line 100 & 102; Ref.23 & 24 are inappropriate because these articles do not mention about CXCR6 and CXCL16.
Line 123; I think that Ref.36 (Lynch et al. ) is more suitable for the sentence (Line 129-133)
Line 140-148; Appropriate references about the role of adipocytes should be added in these sentences.
Line 68-148; I strongly recommend that the authors check the numbering of references in the section 2.
Line 149; I recommend that the authors enlarge the illustration in the quadrilateral of Figure 1.
Line 267; [60.] >>> [60].
Line 270; [51, 69, 60] >>> [51,59, 60]
Line 286; Please add appropriate references for this sentence.
For readers’ understanding, the authors should describe more information about mode of actions of the bone-targeted agents referred in the section 3 and 4.
Author Response
Reviewer #1
Skeletal related events (SREs) such as spinal cord compression and pathologic fractures often occur in men with prostate cancer. Main cause of SREs are treatment-related osteoporosis and bone metastasis, and SREs are correlated to a a decreased QOL and shorted survival in prostate cancer patients. Although bone-targeted agents (BTAs) including bisphosphonates and the antibody against RANKL are approved for the prevention of SREs, it is unclear whether the treatment with BTAs result in prolonged survival. In this article, the authors review the studies on the effectiveness of BTAs in metastatic prostate cancer.
Line 36; Ref.1 may be inappropriate because this article provides the cancer statics in the United States.
Line 64; Please add appropriate references for this sentence (Line 62-64).
Line 80; I think that Ref.15 is suitable for the next sentence (Line 80-82). Please add appropriate references for the sentence (Line 78-80).
Line 100 & 102; Ref.23 & 24 are inappropriate because these articles do not mention about CXCR6 and CXCL16.
Line 123; I think that Ref.36 (Lynch et al. ) is more suitable for the sentence (Line 129-133)
Line 140-148; Appropriate references about the role of adipocytes should be added in these sentences.
Line 68-148; I strongly recommend that the authors check the numbering of references in the section 2.
Line 149; I recommend that the authors enlarge the illustration in the quadrilateral of Figure 1.
Line 267; [60.] >>> [60].
Line 270; [51, 69, 60] >>> [51,59, 60]
Line 286; Please add appropriate references for this sentence.
For readers’ understanding, the authors should describe more information about mode of actions of the bone-targeted agents referred in the section 3 and 4.
----------------------------------------------------------------------------------------------------------
Reviewer: 1
Dear Reviewer, thank you for your suggestions.
- We modified the reference above, according to your comments (doi: 10.1007/978-3-319-95693-0_1., in green). In particular, we believe this paper by Schatten would be more appropriate, since in this manuscript the author provides a brief overview of prostate cancer, also in terms of statistics of this genitourinary malignancy.
- Thank you for this comment. In the revised manuscript, we added a more suitable reference, as suggested by the Reviewer. In addition, we also reported some details on the mechanism of action of zoledronic acid, as reported in the revision #12
- We modified (green), according to your comments, modifying the references of this section
- We added two references about CXCR6 and CXCL16 (green, ref 23 and 24). More specifically, we added an interesting report by Singh and colleagues (reference number 23), where the authors reported the involvement of CXCR6 and its natural ligand CXCL16 in pathobiology of prostate cancer, with CXCL16 stimulation that was suggested to change cytoskeletal dynamics resulting in enhanced migration, invasion and adhesion to endothelial cells; in particular, this mechanism has been suggested to play an important role in enabling prostate cancer cells to achieve their metastatic goal.
In addition, we also added a study by Kapur and colleagues suggesting that CXCR6-CXCL16 axis could be associated with docetaxel resistance, acting as a counter-defense mechanism. We believe these two studies could represent paradigmatic examples, reporting the important role of CXCR6-CXCL16 in this setting.
- We modified (green). The paper by Lynch and colleagues is now the reference number 40
- We added some references regarding the role of adipocytes in this setting, as suggested
- Thank you for this comment. We checked, as recommended
- We enlarged the figure (Figure 1), as suggested by the Reviewer
- We checked, as required
- We checked
- We added the reference number 63
- Thank you for these comments.
We reported some details regarding the mechanism of action of ZA and denosumab, as suggested (purple). In particular, we briefly reported some details in the introduction, in the paragraphs where we cited zoledronic acid. In addition, we also reported some data on denosumab, highlighting the action of this agent in preventing the maturation of osteoclasts precursors, promoting apoptosis of multinucleated, mature osteoclasts.
Thank you again for the time spent to revise our work.
We hope the revised manuscript will better suit Cancers
Reviewer 2 Report
This paper by Mollica V et al, provide an overview on bone tropism of prostate cancer and on the role of bone-targeted agents in metastatic hormone-sensitive and castration-resistant prostate cancer. The manuscript is straightforward, well written, and concise, and has clear results within the scope of a review article. Definitely deserves to be published and is a valuable contribution to the “cancers” journal. Some minor flaws need to be addressed before publication.
Minor points:
[1] “1. Introduction”, Page 1/11, Lines 36-40:
“The natural history of PCa may potentially go through different phases, from a hormone-sensitive prostate cancer (HSPC) to a castration-resistant prostate cancer (CRPC) state and from localized disease to nonmetastatic biochemical recurrence and, finally, to the presence of distant metastases.”
At that point, please, do clarify that prostate cancer cells become resistant to the androgen deprivation therapy (castration-resistant), usually after 18–24 months.
Recommended reference: Saxby H, et al. An Update on the Prognostic and Predictive Serum Biomarkers in Metastatic Prostate Cancer. Diagnostics (Basel). 2020;10(8):549.
[2] “1. Introduction”, Page 2/11, Lines 58-59:
“SREs include pathologic fracture, spinal cord compression, need for surgery or radiation therapy to bone, hypercalcemia [12].”.
Specifically, spinal cord compression represents an oncologic emergency. It occurs in up to 5% of all patients with cancer; however, it is a feature of advanced cancer, most commonly seen in patients with cancers of the breast, lung and prostate, which comprise 60% of cases. Local management strategies generally include palliative RT, or surgical posterior decompression with or without instrumentation or total en bloc spondylectomy.
Recommended reference: Boussios S, et al. Metastatic Spinal Cord Compression: Unraveling the Diagnostic and Therapeutic Challenges. Anticancer Res. 2018;38(9):4987-4997.
[3] “2. Bone metastasis in prostate cancer: the underlying molecular mechanisms”, Page 2/11 until 4/11:
Apart from figure 1, the role of the TGF-β signaling axis should be discussed in this section. Importantly TGF-β enhances the expression of osteoprotegerin, which inhibits osteoclastogenesis. Coincidentally, the activation of TGF-β also promotes the development of bone metastases via stimulating metastatic tumor cells within bone microenvironment to secrete factors that result in osteolytic destruction of bone.
Recommended reference: Mishra S, et al. Blockade of transforming growth factor-beta (TGFβ) signaling inhibits osteoblastic tumorigenesis by a novel human prostate cancer cell line. Prostate. 2011;71(13):1441-54.
[4] General comment (1):
I would also recommend to discuss the inflammation, taken that it is linked with the tumor microenvironment. Findings from available evidence suggest the alleviation of chronic inflammation as a potential therapeutic approach for prostate cancer bone metastases.
Recommended reference: Lu Y, et al. Activation of MCP-1/CCR2 axis promotes prostate cancer growth in bone. Clin Exp Metastasis. 2009;26(2):161-9.
[5] General comment (2):
A workflow diagram for the review would be of benefit for the readers.
Author Response
Reviewer #2
This paper by Mollica V et al, provide an overview on bone tropism of prostate cancer and on the role of bone-targeted agents in metastatic hormone-sensitive and castration-resistant prostate cancer. The manuscript is straightforward, well written, and concise, and has clear results within the scope of a review article. Definitely deserves to be published and is a valuable contribution to the “cancers” journal. Some minor flaws need to be addressed before publication.
Minor points:
[1] “1. Introduction”, Page 1/11, Lines 36-40:
“The natural history of PCa may potentially go through different phases, from a hormone-sensitive prostate cancer (HSPC) to a castration-resistant prostate cancer (CRPC) state and from localized disease to nonmetastatic biochemical recurrence and, finally, to the presence of distant metastases.”
At that point, please, do clarify that prostate cancer cells become resistant to the androgen deprivation therapy (castration-resistant), usually after 18–24 months.
Recommended reference: Saxby H, et al. An Update on the Prognostic and Predictive Serum Biomarkers in Metastatic Prostate Cancer. Diagnostics (Basel). 2020;10(8):549.
[2] “1. Introduction”, Page 2/11, Lines 58-59:
“SREs include pathologic fracture, spinal cord compression, need for surgery or radiation therapy to bone, hypercalcemia [12].”.
Specifically, spinal cord compression represents an oncologic emergency. It occurs in up to 5% of all patients with cancer; however, it is a feature of advanced cancer, most commonly seen in patients with cancers of the breast, lung and prostate, which comprise 60% of cases. Local management strategies generally include palliative RT, or surgical posterior decompression with or without instrumentation or total en bloc spondylectomy.
Recommended reference: Boussios S, et al. Metastatic Spinal Cord Compression: Unraveling the Diagnostic and Therapeutic Challenges. Anticancer Res. 2018;38(9):4987-4997.
[3] “2. Bone metastasis in prostate cancer: the underlying molecular mechanisms”, Page 2/11 until 4/11:
Apart from figure 1, the role of the TGF-β signaling axis should be discussed in this section. Importantly TGF-βenhances the expression of osteoprotegerin, which inhibits osteoclastogenesis. Coincidentally, the activation of TGF-β also promotes the development of bone metastases via stimulating metastatic tumor cells within bone microenvironment to secrete factors that result in osteolytic destruction of bone.
Recommended reference: Mishra S, et al. Blockade of transforming growth factor-beta (TGFβ) signaling inhibits osteoblastic tumorigenesis by a novel human prostate cancer cell line. Prostate. 2011;71(13):1441-54.
[4] General comment (1):
I would also recommend to discuss the inflammation, taken that it is linked with the tumor microenvironment. Findings from available evidence suggest the alleviation of chronic inflammation as a potential therapeutic approach for prostate cancer bone metastases.
Recommended reference: Lu Y, et al. Activation of MCP-1/CCR2 axis promotes prostate cancer growth in bone. Clin Exp Metastasis. 2009;26(2):161-9.
[5] General comment (2):
A workflow diagram for the review would be of benefit for the readers.
----------------------------------------------------------------------------------------------------------
Reviewer: 2
Dear Reviewer, we are really thankful for appreciating the relevance of our work. We have responded to all queries and the changes are now incorporated in the revised version.
In the following section, the comments have been responded pointwise and changes have been highlighted in orange and purple, as specified.
- Thank you for this comment. We reported more data on the onset of castration-resistant prostate cancer, also reporting the required reference (orange)
- We added the study by Boussios and colleagues, as suggested
- We added the reference above (orange), as required
- We added the reference (orange) indicated by the Reviewer, briefly reporting the possible role of inflammation and the association between inflammation and tumor microenvironment.
- Thank you for your suggestion. However, given the current state of art of these treatments in this setting, we would prefer just to keep the image of the preclinical evidence on the “vicious” cycle of bone metastases and the two tables with clinical studies, avoiding a workflow diagram that could appear as a sort of guideline recommendation.
Thank you again for the time spent to revise our work and for your comments.
We hope the revised manuscript will better suit Cancers.